# Assessment of Impact Energy Harvesting in Composite Beams with Piezoelectric Transducers

**DOI:** 10.3390/s21227445

**Published:** 2021-11-09

**Authors:** Nikolaos Margelis, Theofanis S. Plagianakos, Panagiotis Karydis-Karandreas, Evangelos G. Papadopoulos

**Affiliations:** Control Systems Lab, School of Mechanical Engineering, National Technical University of Athens, 15780 Athens, Greece; nmargelis@gmail.com (N.M.); panosgary@gmail.com (P.K.-K.); egpapado@central.ntua.gr (E.G.P.)

**Keywords:** composites, piezoelectric energy harvesting, impact response, impact testing machine

## Abstract

Piezoelectric energy harvesting (PEH) is studied in the case of a low-velocity impact of a rigid mass on a composite beam. A methodology is outlined, encompassing modelling of the open-circuit impact response in a finite element (FE) package, formulation of a lumped parameter (LP) model for the piezoelectric transducer connected with the harvesting circuit, and experimental verification of the impact using a custom portable configuration with impactor motion control. The subcircuit capacitor charging effect, the impactor mass and velocity on the harvesting subcircuit response, and the obtained output power are quantified. The results indicate that the current methodology can be used as a design tool for the structure and the harvesting circuit to achieve power output from composite beams with piezoelectric patches under impact conditions.

## 1. Introduction

Piezoelectric energy harvesting (PEH) is defined as the principle of converting mechanical to electric energy via vibrating piezoelectric transducers, and storing that energy by means of an appropriately designed electric circuit. This process evolves among two main stages: (i) conversion of mechanical strain, stemming from vibration to electric voltage in the piezoelectric (PE) transducer; and (ii) driving the produced current to an electric circuit feeding a battery. Based on this process, pioneering autonomous systems including piezoelectric sensors and appropriate harvesting circuitries have been designed since the late 1990s [1,2]. Most PEH configurations developed since then have been designed by considering as input continuous forced harmonic excitation [3,4]. This led to a variety of devices using ambient or human-motion-induced energy for IoT applications and wireless sensor networks [5,6]. The energy conversion process physics has been studied extensively, employing lumped-parameter models or models based on analytical continuum mechanics for capturing the response of the vibrating structure in the linear [7,8,9] and nonlinear [10,11,12,13,14] regime. A key part of a harvesting device is the circuit. Great effort has been placed on the design of electric circuits maximizing the power output of the PEH device, as in the pioneering works of Ottman et al. [15,16], Lefeuvre et al. [17,18] and others [19,20,21,22,23], leading to various options for the design of a PEH device working under predefined conditions [24,25]. The basic architecture of a PEH circuit has been commercialized [26,27].

An interesting application of PEH is low-energy impact. Neglecting energy loss during an impact, the initial kinetic energy of the impactor is partially converted to potential energy of impactor–target contact, kinetic and elastic energy in the mechanical substructure, while the rest is instantly stored as electric energy in the piezoelectric transducer. The temporal variation of this energy equilibrium may be quantified by means of finite element methods [28], used to model the impacted structure as a continuum solid and encompass the interaction between impactor and structure by means of a contact law. In initial PEH systems, impact was exploited as a frequency up-conversion mechanism, enabling power harvesting at low excitation frequencies. Umeda et al. [29] studied the impact of a steel ball on a piezoelectric vibrator by developing an electrical equivalent model with constants obtained from the admittance characteristics of the system measurements. A plucking-based up-conversion strategy for knee-joint energy harvesting has been developed by Pozzi and Zhu, who modelled a resistively shunted piezoelectric bimorph with finite elements and validated their model experimentally [30]. In several works, the use of an impact as a source of power has been studied by connecting the piezoelectric transducer to a resistance. Renaud et al. [31] developed a lumped-parameter model, assumed a single-mode impulse excitation based on a method reported by Lee [32] and provided a closed expression of electric resistance for optimizing power output. Gu and Livermore studied an impact-driven coupled vibration harvester consisting of a composite and a piezoelectric bimorph beam, by assuming Euler–Bernoulli-type kinematics and displacement at the contact point induced by inelastic collision [33]. The effect of impactor mass and velocity has been studied by Basari et al., based on fundamental energy equilibrium equations and impact tests [34]. Doria et al. [35] studied the effect of the shape and duration of a finite-width impulse of base acceleration on a cantilever beam at open-circuit configuration using analytical, experimental and numerical methods. The contact stiffness between the impactor and the target has been considered by Jaquelin et al. [36] and Martinez-Ayuso et al. [37], who developed analytical solutions based on Euler–Bernoulli beams. Fu and Liao implemented a Hunt–Crossley [38] contact model for predicting the power output of a resistively shunted piezoelectric beam subjected to impact [39]. Circuits including rectifiers have been implemented by Ferrari et al. [40] and Wong and Dahari [41], who studied impacted piezoelectric beams experimentally, in the case of impact induced by a vibrating beam with tip mass and raindrops, respectively. PEH harvesting systems based on impact for powering devices have been developed by Feng et al. [42], who introduced the impact event as an electric pulse excitation in the case of a self-monitoring helmet, and by Chen et al. [43] in the case of road speed-bumps.

In most of the above-mentioned works on PEH, impact has been studied as a frequency up-conversion mechanism, induced as an impulse excitation. However, impulse excitation—mostly induced to the model as a half sine force or voltage time signal—should fit the induced signal amplitude to a measurement. From a design perspective, the velocity of the impactor would be easier to estimate as an analysis input parameter. Moreover, impulse excitation does not encompass the physics of the impact, which is introduced by the contact law between the impactor and the structure hosting the PEH circuit, and mainly depends on the stiffness and mass of the impactor and target. Thus, important parameters such as the impact-force–time profile determining the stress distribution in the impacted structure are not captured accurately. Concerning the circuit, mainly resistive circuits are connected to the structure under impact, leading to a qualitative and not quantitative prediction of the harvested energy. This work aims to complement the current state of the art by presenting a finite element (FE)-based methodology capable of predicting the impact dynamics, including force–time history, and the harvested energy in customized commercial PEH circuits comprised of rectification, charging and discharging phases. The methodology is validated against measurements performed in an in-house low-cost impact testing machine and processed using real-time hardware/software. The results validate the expected harvested power, and the effect of impact parameters such as impactor mass and velocity on the coupled electromechanical response.

## 2. Description of the Method

The objective of the experimental-numerical method developed here is the prediction of the electric power that can be harvested during impacts which start vibrations on composite beams with piezoelectric patches connected to PEH circuits. The methodology evolves in two stages, indicated by the orange and the blue dotted lines in the flowchart shown in Figure 1.

In the first stage, a finite element (FE) model of a composite beam with a piezoelectric patch is developed in the multiphysics commercial software Abaqus, where the patch is in open-circuit mode [44]. This stage serves to validate the numerical model of the beam subject to impact tests. The validation leads to fine-tuning of the FE model in terms of contact stiffness and energy dissipation on the composite material and its support. In this context, the methodology is described as experimental-numerical, since some of the model parameters (e.g., contact stiffness and damping) are experimentally determined. Next, the selected terms of modal matrices and the predicted slope of the FE model are used as input to the analytical equations for predicting the current in closed-circuit mode [9]. In this context, the piezoelectric patch is modelled as an electromechanically uncoupled current source. Then, the structure of the PEH circuit is programmed in a commercial circuit-design software [45] to yield the transient and steady-state responses of the electromechanical system. The predictions of the closed electric circuit model are compared with measurements.

In the next sections, the basic equations of motion of the coupled electromechanical system are formulated, and each stage of the methodology is described explicitly.

### 2.1. Coupled Electromechanical System in Open-Circuit Conditions

The beam geometry is shown in Figure 2, and the relevant geometric parameters are listed in Table 1. The PE patch, consisting of a piezoceramic material embedded in polyimide, is bonded with epoxy on the upper surface of the beam and is assumed to be tightly connected with the beam. The beam laminate consists of orthotropic composite plies with arbitrary orientation and an isotropic piezoelectric transducer. The composite material is assumed to exhibit linear elastic behavior. Through-thickness polarization and linear piezoelectric response are assumed for the PE patch transducer.

The ply constitutive equations in the natural coordinate system Oxyz (Figure 2) are [46]:(1)$$\begin{array}{c} \sigma_{i}=C_{\mathrm{ij}}^{E}S_{j}-{\left(e_{3i}\right)}^{T}E_{3}\\ D_{3}=e_{3j}S_{j}+\varepsilon_{33}^{S}E_{3} \end{array}$$
where i, j = 1,…,6; σ_i_ and S_j_ are the mechanical stress and engineering strain in vectorial notation; E_3_ is the component of electric field vector along the thickness direction; D_3_ is the electric displacement vector component; C_ij_ is the elastic stiffness tensor; e_3j_ is the piezoelectric tensor arising from the piezoelectric charge tensor and the stiffness tensor and ε_33_ is the material electric permittivity. Superscripts E and S indicate constant electric field and strain conditions, respectively, and superscript T indicates matrix transposition. Equation (1) is formulated in a general manner to encompass the behavior of both a piezoelectric and a passive composite ply (e_mj_ = 0). The electric field vector component E_3_ is the gradient of the electric potential V along the thickness of the PE patch:(2)$$E_{3}=-\partial V/\partial z$$

The through-thickness mechanical displacements of the composite laminate are described employing the first-order shear laminate theory:(3)$$\begin{array}{c} u\left(x,y,z\right)=u^{0}\left(x,y\right)+\beta_{x}\left(x,y\right)z\\ v\left(x,y,z\right)=v^{0}\left(x,y\right)+\beta_{y}\left(x,y\right)z\\ w\left(x,y,z\right)=w^{0}\left(x,y\right) \end{array}$$
where u^0^, v^0^, w^0^ and β_x_, β_y_ are displacements and rotations of the midsurface, respectively. As explicitly described in Section 2.2, a higher-order layerwise approximation of the through-thickness displacement field [47] is utilized at a specific stage of the method to directly provide the slope of transverse displacement along the longitudinal axis of the beam. The PE patch is modelled as a 3D solid continuum, perfectly bonded on the composite substrate, with electric potential as nodal DOF in addition to mechanical displacements.

The equations of motion of the beam subjected to impact are expressed in variational form using Hamilton’s principle as [28]:(4)$$\int_{t_{0}}^{t_{0}+\mathrm{dt}}\left(\int_{V}\delta KdV-\int_{V}\delta HdV-\int_{V}{\delta W}_{d}dV+\int_{\Gamma }\delta \bar{u}{}^{T}\bar{\tau }d\Gamma \right)\mathrm{dt}=0$$
where index *V* denotes beam volume; $\delta \bar{u}$ is the vector of mechanical displacements arising from the kinematic assumptions (3); $\bar{\tau }$ are the tractions at the boundary surface Γ and δH, δK and $\delta W_{d}$ are the variations of the electromechanical, kinetic, and dissipated energy of the beam, respectively. The energy terms in Equation (4) may be expressed as a function of stress, strain and electric field, which are substituted using Equations (1) and (2) and the strain–displacement relations. The degrees of freedom of the beam are approximated along the midsurface (mechanical displacements) and within the transducer volume (electric voltage) by means of quadratic (8 nodes/FE) and linear (4 nodes/FE) shape functions, respectively [44]. Numerical integration of Equation (4) yields the coupled electromechanical discrete system in the time domain:(5)$$\left[\begin{array}{cc} \left[M_{\mathrm{uu}}\right] & 0\\ 0 & 0 \end{array}\right]\left\{\begin{array}{c} \bar{\ddot{u}}\\ \ddot{V} \end{array}\right\}+\left[\begin{array}{cc} \left[C_{\mathrm{uu}}\right] & 0\\ 0 & 0 \end{array}\right]\left\{\begin{array}{c} \bar{\dot{u}}\\ \dot{V} \end{array}\right\}+\left[\begin{array}{cc} \left[K_{\mathrm{uu}}\right] & \left[K_{u\upsilon }\right]\\ \left[K_{\upsilon u}\right] & \left[K_{\upsilon \upsilon }\right] \end{array}\right]\left\{\begin{array}{c} \bar{u}\\ V \end{array}\right\}=\left\{\begin{array}{c} F\left(t\right)\\ Q\left(t\right) \end{array}\right\}$$
where [M], [C] and [K] are the system mass, damping and stiffness matrices, divided into elastic (index u) and electric (index υ) parts; F is the external force vector and Q denotes the external electric charge. In the case of an open-circuited piezoelectric transducer, as in the first stage of the methodology illustrated in Figure 1, the external electric charge is assumed to be zero.

The external force appearing in Equation (5) is attributed to the contact between a rigid impactor, modelled as a point mass with initial velocity, and the beam. This impact force is expressed as a function of transverse displacement using a linear contact law:(6)$$F\left(t\right)=\left\{\begin{array}{cc} k_{y}\left(w^{i}\left(t\right)-w^{0}\left(x_{0},y_{0},t\right)\right), & w^{i}\left(t\right)>w^{0}\left(x_{0},y_{0},t\right)\\ 0 & w^{i}\left(t\right)\leq w^{0}\left(x_{0},y_{0},t\right) \end{array}\right\}$$
where the index i denotes the impactor, (x_0_, y_0_) are the coordinates of impact on the midsurface and k_y_ is the contact stiffness derived by material and geometric properties [48]. The contact law of Equation (6) is represented schematically by interposition of a linear spring with stiffness k_y_ between beam and impactor, as shown in Figure 3.

Note that the present formulation accounts for low-velocity impacts causing no material damage. It was implemented using the linear spring connection element SPRINGA [44]. More sophisticated models describing the contact between impactor and composite beam [49,50,51] could be applied, but the linear contact law was used for the sake of simplicity.

Solution of Equation (5) provides the time profile of displacements of the composite beam and the electric potential at the terminals of the open-circuited piezoelectric transducer, which in this case acts as a sensor. The time profile of impact force may be predicted using Equation (6) and used as input for predicting electric current in a harvesting circuit, as explicitly described in the following section.

### 2.2. Determination of the PEH Patch Model

The response of the PEH patch is predicted by developing a model with degrees of freedom depending on the vibration modes considered, which in the single-mode case is described as a lumped parameter (LP) model with parameters determined by the FE simulation. The model is based on the formulation of Erturk and Inman [9]—the piezoelectric patch is modelled as a current source in parallel to an electric impedance (Figure 4) and a capacitor, which is practically the capacitive part of the piezoelectric transducer.

The closed-circuit response is described by the ordinary differential equation [9]:(7)$$\frac{{\bar{\varepsilon }}_{33}^{S}\cdot b\cdot L}{h_{p}}\cdot \frac{\mathrm{dV}}{\mathrm{dt}}+\frac{V\left(t\right)}{R_{l}}+\overset{\infty }{\underset{j=1}{\sum }}\kappa_{j}\frac{{d\eta }_{j}\left(t\right)}{\mathrm{dt}}=0$$

The capacity in the circuit of Figure 4 is:(8)$$C_{p}=\frac{\varepsilon_{33}\mathrm{bL}}{h_{p}}$$
where L, b and h_p_ are the length, width and thickness of the transducer, respectively.

Considering that an exclusively bending vibration along the length of the beam is excited by the impact event and assuming proportional damping, the transverse displacement is separated to spatial and time components:(9)$$w\left(x,t\right)=\overset{n}{\underset{j=1}{\sum }}\varphi_{j}\left(x\right)\eta_{j}\left(t\right)$$
where j denotes an eigenmode, n is the number of the eigenmodes considered and φ is an eigenvector. The circuit current source is derived by comparing the analytical bending equation of motion with the dynamics of a resistor–capacitor circuit, as:(10)$$I\left(t\right)=-\overset{n}{\underset{j=1}{\sum }}\kappa_{j}\frac{{d\eta }_{j}\left(t\right)}{\mathrm{dt}}$$
where j denotes an eigenmode and η is the transient vector of modal coordinates in the space-time decomposition of the transverse displacement. κ is the electromechanical coupling coefficient, provided as:(11)$$\kappa_{j}=e_{31}h_{p}b\int_{x_{s}}^{x_{f}}\frac{d^{2}\varphi_{j}\left(x\right)}{{\mathrm{dx}}^{2}}\mathrm{dx}=e_{31}h_{p}b\left({\left.\frac{{d\varphi }_{j}\left(x\right)}{\mathrm{dx}}\right|}_{x_{f}}-{\left.\frac{{d\varphi }_{j}\left(x\right)}{\mathrm{dx}}\right|}_{x_{s}}\right)$$
with x_s_ and x_f_ denoting the start and end of the patch in terms of length coordinate x.

The methodology followed for determining stiffness, mass, damping, piezoelectric and permittivity terms of the LP model, including all necessary terms appearing in Equations (9)–(11), may be briefly summarized as follows.

(a)Solving of Equation (5) for the free-vibration response of the open-circuit system in the frequency domain provides the eigenvectors, which are used for the formulation of the modal mass, damping, stiffness and force transformation matrices, respectively [52]:
(12)$$\begin{array}{c} \left[M_{\mathrm{ffmod}}\right]={\left[\varphi \right]}^{Τ}\left[M\right]\left[\varphi \right]\\ \left[C_{\mathrm{ffmod}}\right]={\left[\varphi \right]}^{Τ}\left[C\right]\left[\varphi \right]\\ \left[K_{\mathrm{ffmod}}\right]={\left[\varphi \right]}^{Τ}\left[K\right]\left[\varphi \right]\\ \left[F_{\mathrm{sfmod}}\right]={\left[\varphi \right]}^{Τ}\left[F\right] \end{array}$$(b)The modal matrices are used for determining η:(13)$$\left[M_{\mathrm{ffmod}}\right]\left\{\ddot{\eta }\right\}+\left[C_{\mathrm{ffmod}}\right]\left\{\dot{\eta }\right\}+\left[K_{\mathrm{ffmod}}\right]\left\{\eta \right\}=\left\{F_{\mathrm{sfmod}}\right\}u$$
where *u* is a modal force amplitude scaling parameter that can be predicted by Equations (5) and (6) or obtained by measurement of the impact force. The linear system of Equation (13) consists of n uncoupled equations.(c)The electromechanical coupling coefficient κ is provided by Equation (11) using PE properties, geometrical parameters and the slope of the modal transverse displacement. The latter is directly calculated in modal space, in addition to the eigenvectors and modal matrices, by implementing a C^1^-continuous 2D higher-order layerwise FE [47], which encompasses this slope as a nodal DOF.(d)The electric current flowing through the PEH circuit is calculated using Equation (10). As an alternative to points (b–c), η may be derived at the point of excitation by means of a high-speed camera.(e)The PE transducer is modelled in the PEH circuit as a current source and a capacitor with C_p_ appearing in Equation (8).(f)The harvested power may be calculated as:(14)$$P\left(t\right)=\frac{\int_{t_{1}}^{t_{2}}V_{\mathrm{out}}\left(t\right)I_{\mathrm{out}}\left(t\right)\mathrm{dt}}{t_{2}-t_{1}}$$
where V_out_ and I_out_ are respectively the voltage and current in the discharging branch of the PEH circuit described in the following section.

### 2.3. PEH Circuit

The PEH circuit studied here is the commercial PI E-821.00 [27], shown in Figure 5a. The circuit was modelled in Ltspice software [45], as presented in Figure 5b. The model includes a charging branch, consisting of a diode rectifier and a capacitor that charges due to the alternative current produced by the PE transducer. When the voltage of the capacitor reaches a reference value (nominal value 12 V), an automatic switch is activated, allowing part of the electric energy in the capacitor to be released into the discharging branch. The automatic switch is turned off when the voltage of the capacitor falls below a low threshold (nominal value 6 V). The switch is implemented by a MOSFET controlled by an operational amplifier which compares the voltage of the capacitor to the reference voltage. This response is modelled with the use of a Schmitt trigger supplied by external voltage sources for simplicity. The discharging branch consists of a buck converter tuning the output to be a pulse of constant voltage.

## 3. Experimental Configuration for Impact Testing

### 3.1. Custom Impact Frame for Controlled Impact Velocity

Impact tests were performed using a custom portable experimental configuration [53], which was complemented in terms of repeatability and data acquisition capabilities (Figure 6).

Its basic version includes a light aluminum frame, prestressed by steel cables for providing extra rigidity, and an impactor instrumented for impact force measurement. In its current version, the impactor is moved by a DC servomotor [54] in controllable configuration by means of encoder feedback (see Appendix A). The perpendicularity of impact is ensured by means of two parallel bearing sets. Impact force and impactor velocity are measured by a custom force sensor [55] and the encoder, respectively.

The impact test takes place in two successive steps: (i) the impactor is raised up to the desired height, and (ii) the velocity of the impactor is provided by the user to initiate its trajectory (see Appendix A). The motion of the impactor in both stages is controlled by implementing a modified state feedback controller [56] with gravity compensation. The controller is based on a lumped-parameter model of the impactor, including constants derived by CAD models and static, constant velocity and constant acceleration tests (see Appendix A). The appropriate gains were determined by the root-locus method. All signals (load cell, encoder feedback, circuit voltage and current) are obtained using high-speed data acquisition modular equipment (NI CRIO 9074 [57] including an analog input module (AI–NI 9220) with 16 channels being capable to operate at a maximum rate of 10^5^ samples/s per channel, and a digital I/O module (NI 9401) with a maximum sampling rate of 10^7^ samples/s, while a rate of 10^4^ samples/s is used for the control of the impactor. The software implemented for acquisition, control and storage of the signals is Labview FPGA [58]. The developed experimental configuration [59] enables impactor tip angle and velocity to be set by the test engineer to perform impact tests of various types [60] (see Appendix A). The displacement at the load application point is measured by processing images [61] acquired by a high-speed camera. A polyurethane tip was used in the impact tests. The positions of impactor and piezoelectric patch on the tested clamped composite beam are shown in Figure 2.

### 3.2. Materials and Specimens

A composite beam of graphite/epoxy material with a P.876-A12 DuraAct^®^ [62] (PI Ceramic GmbH, Lederhose, Germany) surface attached piezoelectric patch and lamination [−45/45/90_2_/0/90]_S_ was studied. The patch consists of PIC255 piezoceramic embedded in polyimide. For the sake of simplicity, the piezoceramic part of the patch was modelled, and its properties are listed in Table 2. The elastic properties of the composite material considered were obtained from static tests [63] performed by the manufacturer (Hellenic Aerospace Industry S.A., Athens, Greeece). They were validated additionally by performing modal tests on the cantilever beam leading to measurement of the fundamental eigenfrequency at 89.7 Hz. A relatively large modal loss factor of η = 12.5% was assumed in the LP system, which includes the contribution of clamped support, viscoelastic material and intensive usage of the specimen in forced harmonic response tests [64]. The electromechanical properties of all materials considered are listed in Table 2.

## 4. Results and Discussion

According to the methodology shown in Figure 1, the validation stage included the study of the composite beam impact response in open-circuit electric conditions; therefore, the piezoelectric transducer acted as a sensor. The contact stiffness k_y_ was tuned by correlating model prediction with measured impact force and piezoelectric voltage time profiles, yielding a value of 500 N/m. In the prediction stage, the closed-circuit response was studied in three cases, each with a different electric component connected to the PE terminals: (i) a resistance of 10 kΩ; and (ii) the E-821.00 harvesting module twice, each time with a different charging subcircuit capacitor. The latter configuration was achieved by replacing the standard 200 μF capacitor with either of two other custom ones having a capacity of 100 μF and 1 nF, respectively.

### 4.1. Open-Circuit Impact Response

Figure 7 illustrates FE-predicted and measured time history of impact force, transverse displacement, and piezoelectric voltage at an impact with initial velocity of 0.75 m/s. The measured force signal shown in Figure 7a indicates a globally dominated response [53], as derived from the long duration of the impact and the single fundamental vibration mode participating in the response. Chattering was observed in the beginning of the measured force signal, assumed to be mainly attributable to the dimensional clearance of the force sensor in the impactor assembly. The FE model successfully captured the impact duration, while it overestimated the impact force amplitude. This deviation may be attributed to modelling the impactor as a concentrated rigid mass, whereas in the test the polyurethane tip became deformed during impact. A better comparison during the initial impact phase was observed between prediction and measurement in the time history of the displacement, shown in Figure 7b. The measured piezoelectric voltage (Figure 7c) had a similar time profile with the displacement, as expected in an open-circuit PE response, while it was more sensitive to the initial chattering observed in the force–time history. The end of the impact event was determined from the force sensor signal, considering the initial noise, and is denoted with a dashed vertical line.

### 4.2. Closed-Circuit Resistive Impact Response

Predictions of the lumped parameter (LP) model for the resistor voltage and circuit current are compared with measurements in Figure 8a,b, respectively. The LP model predictions followed the measured trends for both the resistor voltage and circuit current, but missed the fluctuation measured during the initial impact phase. The change of sign in both graphs during impact indicates that the impactor moved backwards when it lost contact with the beam. The free vibration which followed the impact event, provided a smoother signal, which contained practically the first bending mode.

### 4.3. Closed-Circuit PEH Response

The dynamic response of the PEH circuit connected to the piezoelectric composite beam under an impact with velocity of 0.75 m/s is illustrated in Figure 9 and Figure 10. A 100 μF capacitor was employed in the circuit. Voltage and current were measured at the piezoelectric terminals and at the capacitor, as illustrated in Figure 5b. The current predictions and measurements seemed to be more sensitive to the impact event and the induced vibration, while the voltage appeared to have a smoother time response. This trend, also observed in the forced frequency response of a similar system [64], is more evident in the measurements presented in Figure 10. This may be attributed to the rectification in the charging subcircuit, which mainly affected the voltage signal as designated by the manufacturer. As shown in Figure 10, the capacitor voltage eventually reached a value of 0.22 V, which is lower than the discharging threshold of 12 V, thus there was no current output in the discharging subcircuit.

The LP model could predict the general signal trends, but missed the initial fluctuation during impact. This trend was more evident before rectification (Figure 9). As already observed in the open-circuit case (Section 4.1), parameters not included in the impact modelling approach, such as the deformability of impactor and clearance in the force sensor assembly, strongly affected the fluctuation in the dynamic response during the initial phase of the impact event. Nevertheless, this methodology could predict the limit values of impact-induced signals and therefore could be used as a design tool for PEH circuits in applications related to low-velocity impact events.

The design capabilities of the developed methodology were studied by implementing a 1 nF capacitor in a customized version of the E-821.00 PEH circuit for achieving voltage output. This capacity is smaller than the 2.7 μF capacity of the MOSFETs, so practically the circuit operated with the MOSFETs also acting as capacitors.

Predictions and measurements of output voltage in the case of an impact velocity of 1.25 m/s are shown in Figure 11. A very good match between predicted and measured signal was observed, indicating the enhanced capabilities of the methodology. The low value of capacitance required for obtaining harvested power indicates that the cantilever beam configuration is not the most appropriate for impact harvesting, since the strain in the piezoelectric transducer was not sufficient to charge capacitors of a practical application range in realistic implementation at macro-scale. To this end, other configurations should be further studied, focusing on the design of the composite structure hosting the transducer.

The current methodology could be used as a design tool for harvesting circuits in the case of structures subjected to impact. In case of known impactor mass and velocity, a custom PEH circuit can be designed to harvest power during impact and subsequent vibration. Conversely, for the fixed PEH with a 2.7 μF capacitor studied herein, the mass and velocity of the impactor can be modified, representing different impact conditions.

Figure 12 illustrates the effect of initial impactor velocity on the harvested power. The blue stars indicate experimental data, while the green circles are predictions of the present LP model for an impactor mass of 0.3 kg. For velocities exceeding 1 m/s, the beam acted as a harvester. Power increases had two discrete slopes. However, a precise prediction of the harvested power at high velocities requires a thorough study of the mechanical response of the beam in terms of geometric linearity and mechanical strength, which is beyond the scope of the current work.

In Figure 13, the effect of a mass impacting the beam with initial velocity 0.75 m/s is shown. The threshold for harvesting power in the E.821.00 PEH circuit with a modified capacitor was 0.8 kg. The harvested power increased with mass, as expected due to the increased strain induced in the piezoelectric patch by the larger impulse of the impactor.

In addition to the prediction of harvested power, the current methodology can be used as a design tool for prediction of the structural integrity of a structure subjected to impact. The prediction of the impact force–time profile, as presented in Figure 7, provides an overview of the impact event and the timestamp of maximum impact force. Figure 14 illustrates the longitudinal stress distribution at ply 3 of 12 at maximum impact force in the case of an impact of a 0.3 kg mass with a velocity of 4 m/s. This ply has a 90° fiber orientation with respect to the coordinate system presented in Figure 2. In specific regions, the compressive stress exceeded the compressive strength of the epoxy matrix (158 MPa) [63]. This prediction dictates that an additional design loop is required to ensure the structural integrity of the harvesting system and tune its harvesting capability by properly setting the capacity of the output subcircuit.

## 5. Summary and Conclusions

A FE-based methodology was developed for studying PEH from composite beams with piezoelectric patches subjected to low-velocity impact. Equivalent LP models were implemented for predicting the post-impact free vibration, electromechanical response of the beam, and the impact-induced electric response of the harvesting circuit. A commercial PEH circuit was modified to enable harvested power output under the specific impact conditions achieved using a custom portable experimental configuration, which involves programming in real-time software for controlling impactor velocity and acquiring sensory signals. The main conclusions derived are listed below:Predictions of the current methodology compared well with measurements of the electric response of the harvesting circuit, indicating its applicability in the design of PEH systems subjected to impact.Deviations between predictions and measurements were observed for the impact force due to the geometric clearance of the force sensor, approximation of impactor–beam contact stiffness and impactor deformability.Modification of the commercial PEH circuit in terms of capacity in the charging subcircuit led to harvesting of power in the impact conditions studied experimentally. The design of this modification was possible via the development of a circuit model equivalent to the commercial circuit in LTspice software, and respective verification.The harvested power increased with impactor mass and velocity beyond a threshold. Up to that threshold, no electric power produced due to the impact event and subsequent beam vibration could be extracted.A major capability of this methodology is the prediction of impact-force–time profile and stress distribution during impact events in composite beams with arbitrary lamination. In this context it can be used for the design of composite impact harvesters with piezoelectric patches, enabling the prediction of harvested power within applicable structural integrity limits.

The current methodology includes the modelling of PEH in composite beams by developing an LP electromechanical model based on FE and solving for the electric response of the harvesting circuit. To this end, the extension of the methodology to plate and shell structures is straightforward and will be studied in future work, aiming at the design of appropriate PEH circuits for various impact conditions.

## Figures and Tables

**Figure 1 sensors-21-07445-f001:**
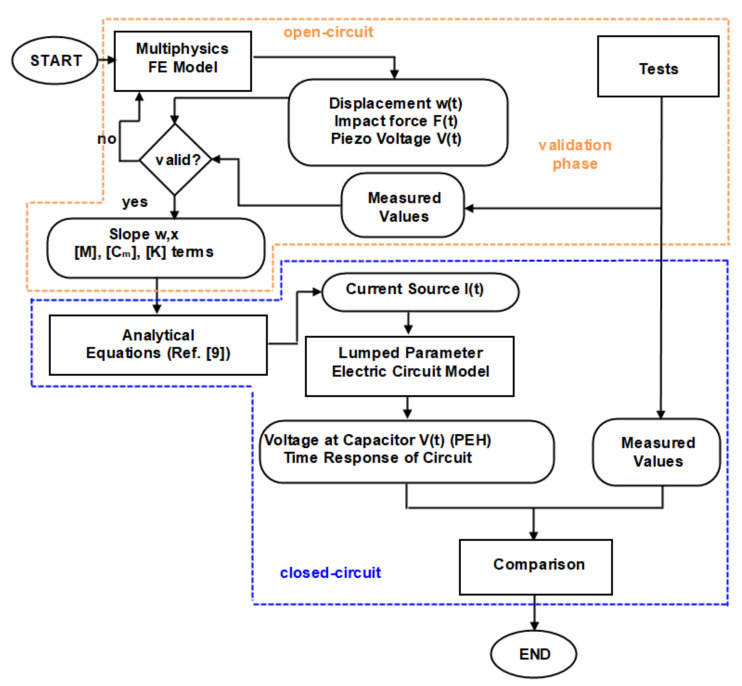
Flowchart of the experimental-numerical methodology.

**Figure 2 sensors-21-07445-f002:**
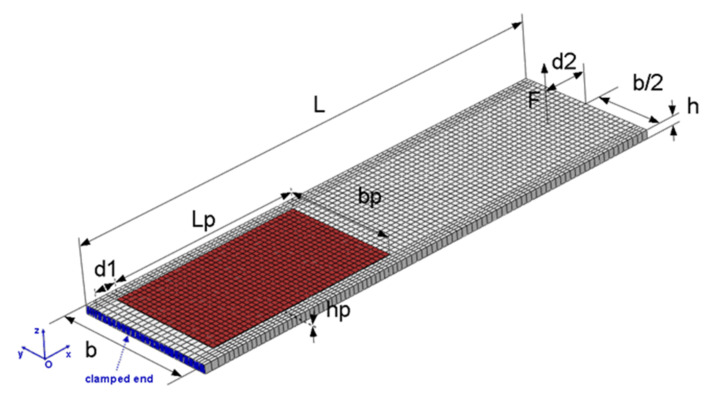
Composite beam with PE patch: FE mesh and geometric parameters.

**Figure 3 sensors-21-07445-f003:**
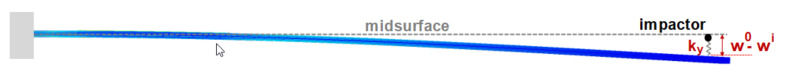
Modeling of impactor–target contact during impact.

**Figure 4 sensors-21-07445-f004:**
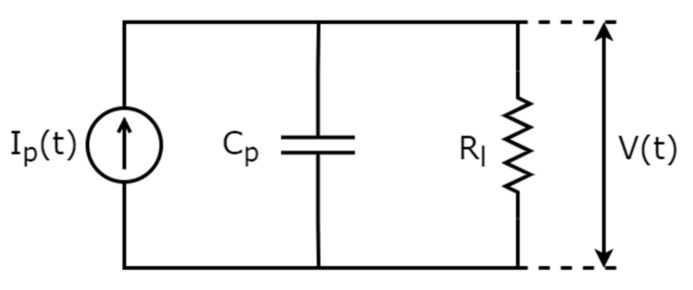
Modeling of piezoelectric patch in closed-circuit configuration.

**Figure 5 sensors-21-07445-f005:**
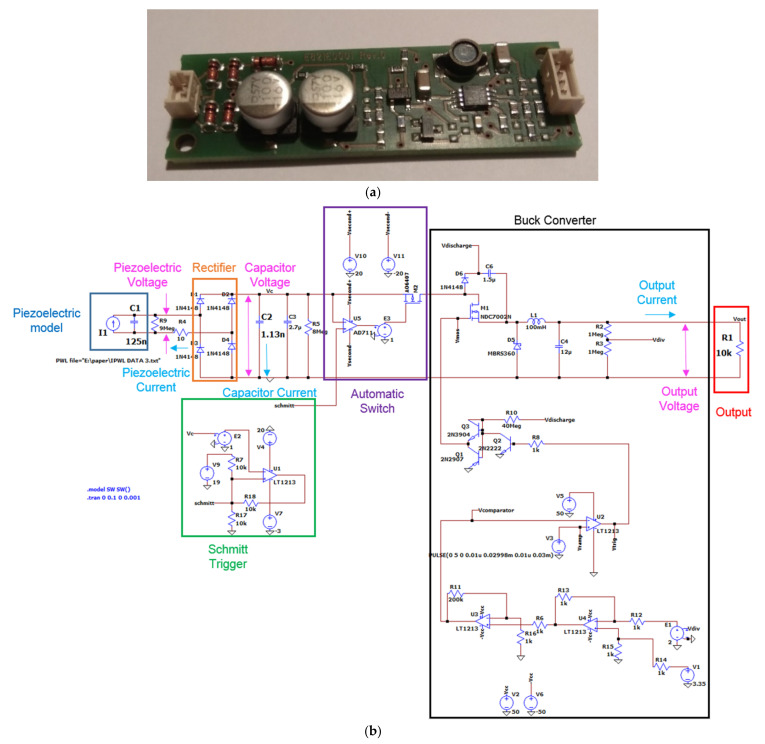
PEH circuit: (**a**) physical item, (**b**) modelling approach in Ltspice.

**Figure 6 sensors-21-07445-f006:**
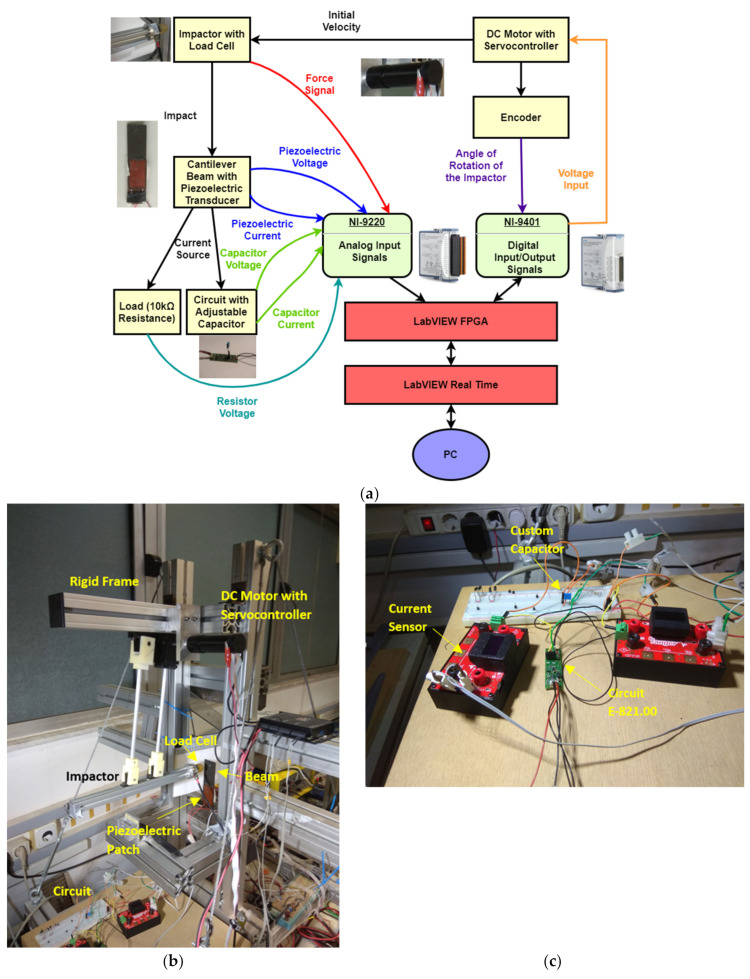
Experimental configuration for impact tests: (**a**) schematic representation with measurement system, (**b**) photo of impact frame with beam specimen, (**c**) PEH circuit including an E-821.00 module.

**Figure 7 sensors-21-07445-f007:**
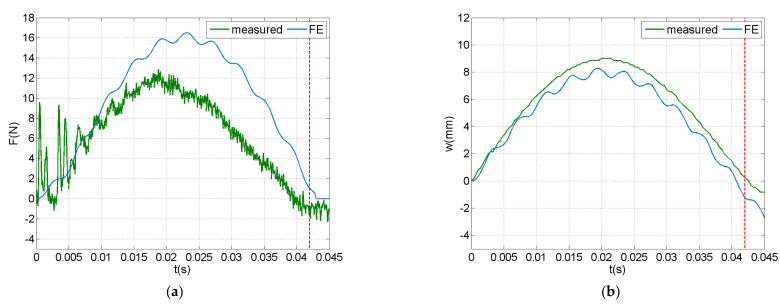
Time history of predicted and measured impact response (v^i^ = 0.75 m/s) in open-circuit configuration: (**a**) impact force, (**b**) tip displacement and (**c**) electric potential at piezoelectric terminals.

**Figure 8 sensors-21-07445-f008:**
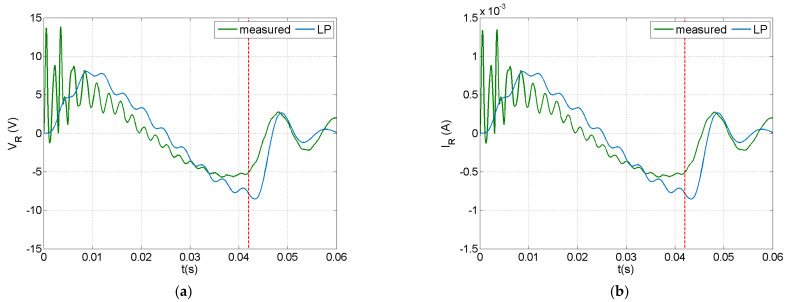
Impact response in a resistive circuit with R = 10 kΩ: (**a**) resistor voltage, (**b**) circuit current.

**Figure 9 sensors-21-07445-f009:**
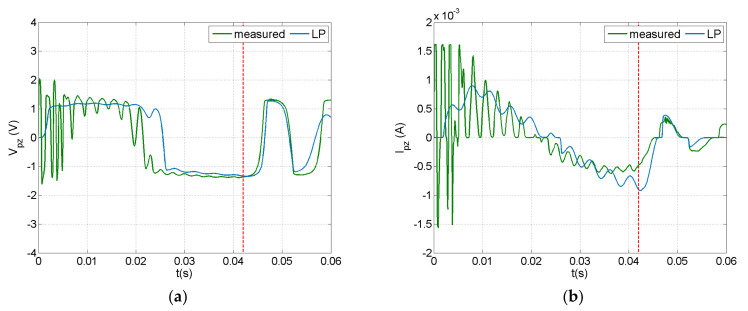
Predicted and measured signals at the piezoelectric terminals of the PEH circuit with a 100 μF capacitor in the case of impact: (**a**) voltage, (**b**) current.

**Figure 10 sensors-21-07445-f010:**
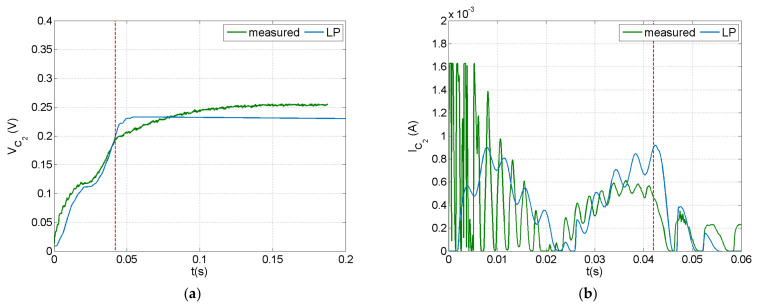
Predicted and measured signals at the capacitor of the PEH circuit with a 100 μF capacitor in the case of impact: (**a**) voltage, (**b**) current.

**Figure 11 sensors-21-07445-f011:**
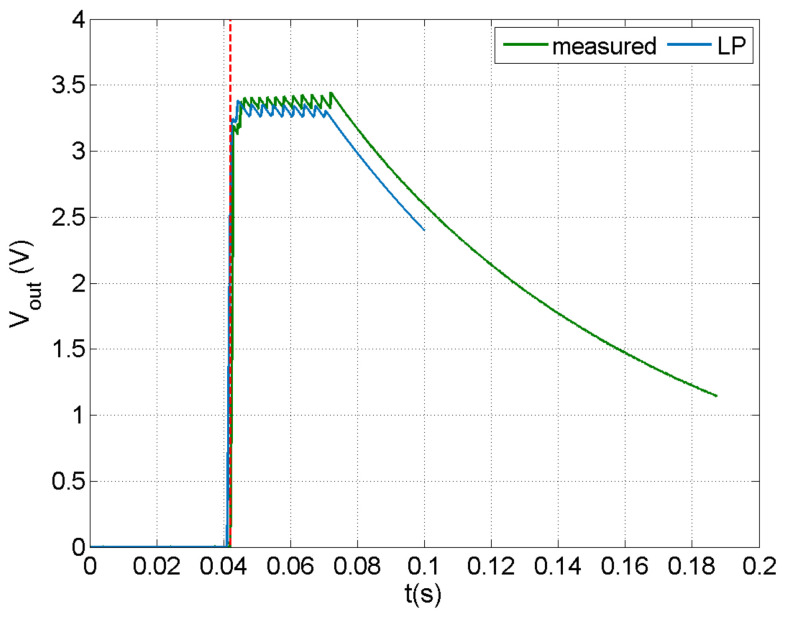
Predicted and measured voltage in the output of the discharging subcircuit in the case of a PEH circuit with capacity of 2.7 μF.

**Figure 12 sensors-21-07445-f012:**
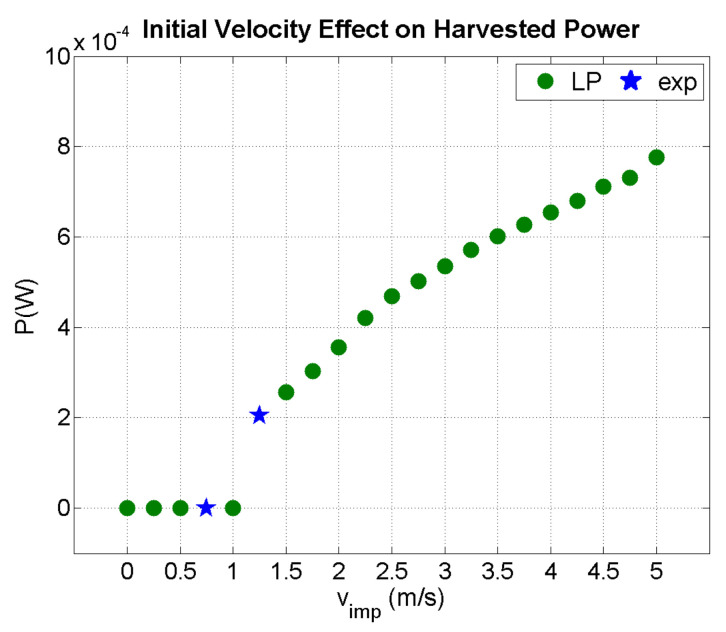
Predicted output power as a function of impact velocity (m^i^ = 0.3 kg).

**Figure 13 sensors-21-07445-f013:**
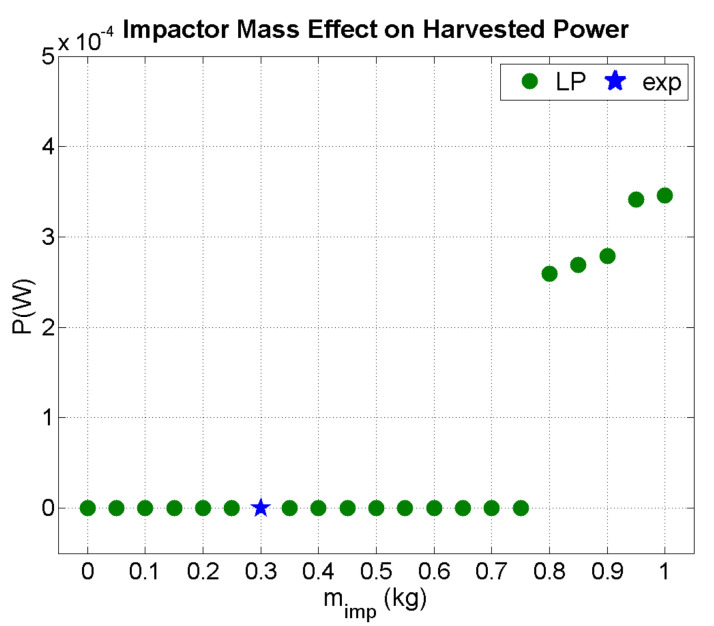
Predicted output power as a function of impactor mass (v^i^ = 0.75 m/s).

**Figure 14 sensors-21-07445-f014:**
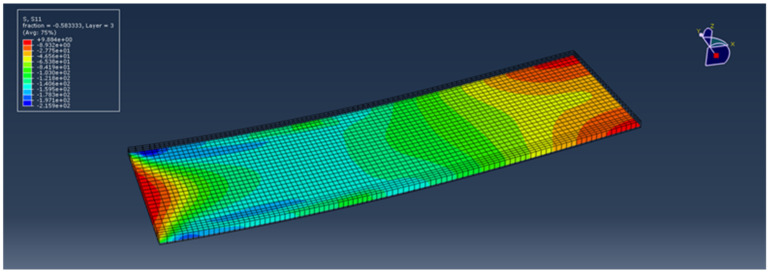
Longitudinal stress distribution in ply 3 of a composite beam at maximum impact force during the impact of a 0.3 kg mass hitting with 4 m/s initial velocity.

**Table 1 sensors-21-07445-t001:** Beam and PE patch geometric parameters.

Parameter	Value (mm)	Parameter	Value (mm)
Beam length L	128	PE patch length L_p_	50
Beam width b	37	PE patch width b_p_	30
Beam thickness h	2.15	PE patch thickness h_p_	0.2
Distance from support d_1_	5.5	Distance from free end d_2_	10

**Table 2 sensors-21-07445-t002:** Electromechanical properties of materials considered (* under constant stress).

Property	Composite Material	Piezoelectric Material
Density (kg/m^3^)	1554	7800
*Elastic Properties*		
E_11_ (GPa)	138.40	62.10
E_22_ (GPa)	8.50	62.10
E_33_ (GPa)	8.50	48.30
G_12_ (GPa)	4.30	23.20
G_13_ (GPa)	4.30	21.30
G_23_ (GPa)	4.30	21.30
ν_12_	0.31	0.33
ν_13_	0.31	0.43
ν_23_	0.31	0.43
*Piezoelectric Properties*		
d_31_ (10^−12^ m/V)	-	−191
d_32_ (10^−12^ m/V)	-	−191
d_33_ (10^−12^ m/V)	-	409
*Dielectric Properties* (ε^0^ = 8.85 × 10^−12^ F/m)		
ε_33_/ε^0^	3.5	1832 *

## Data Availability

Data available upon request from the authors.

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
