# Peer review of "Assessment of Impact Energy Harvesting in Composite Beams with Piezoelectric Transducers"

_sensors, 2021, doi:10.3390/s21227445_

Round 1
Reviewer 1 Report
In my opinion, the manuscript is suitable for publication in Sensors journal but Authors must complete a major revision. Manuscript should be revised according to following comments:
1. Chapter “Introduction” should be improved:
a) Author must decide: what is the literature review about?
- about process of impact energy harvesting,
- or about method of “assessment of impact energy harvesting”.
If method then methods, used by other researcher, should be described. The manuscript only mentions that other researchers have dealt with “impact energy harvesting” but there is no description of their methods.
2. Chapter “Description of the method” must be improved:
a) Author use the term “experimental-numerical methodology”. Why is such term (not used in other articles) used? Many articles use the comparison of the results of numerical and simulation studies. Author should explain the causes,
b) line 89: Author should explain what is the PE patch (PZT?),
c) equation 1: why is sign “-“ in first equation? Standard version of constitutive equation contains sign “+”.
3. Chapter “Experimental Configuration for Impact Testing” must be improved:
a) Author present views of laboratory stand. However, not everything is clear on the basis of these views. Schema of measurement system should be added. In Appendix A control system is above all presented.
4. Chapter “Summary and Conclusion” must be improved:
a) this chapter must be substantially processed, because some of conclusion are obvious,
b) conclusions should fill research gap which is described in introduction.
For example: Authors wrote in introduction: "Thus, important parameters, such as the impact-force time profile determining the stress distribution in the impacted structure, are not captured accurately."
So, what is the influence of the impact-force time profile on the stress distribution?
5. Appendix A must be improved:
a) line 400: what is a PV controller?
b) there is no description of the signs in Figure 2, What is: r, Kp, e, tw, etc?,
c) there is no description of the signs in Figure 3 and 4.
Reviewer 2 Report
The topic is interesting and numerical results are corroborated by experimental results
Specific remarks:
Line 31: Lumped parameter models are a simplification of continuum mechanics, please clarify.
Line 58: The authors should cite other FE studies e.g. DORIA A., C. MEDÈ C., D. DESIDERI D., MASCHIO A., CODECASA L., MORO F., (2018), On the performance of piezoelectric harvesters loaded by finite width impulses, Mechanical Systems and Signal Processing, Vol. 100 (2018) 28–42.
Figure 2: the clamp should be highlighted.
Line 135: The authors should mention more sophisticated contact models, e.g. Skrinjar, L.; Slavic, J.; Boltežar, M. A review of continuous contact-force models in multibody dynamics. Int. J. Mech. Sci. 2018, 145, 171–187
Eq. 9 The authors should highlight that the equation holds in the presence of proportional damping.
Eq. 13. The right-hand side represent modal forces, please clarify.
Line 219: The authors should give the velocity profile of the impactator.
Line 231: Please define tip angle.
Line 243: a modal loss factor of 25% is rather large, the authors should give more details.
Table 2: What piezo material is simulated?
Line 278: What is the value of resistance?
Line 333: The authors should give information about power calculation.
Round 2
Reviewer 1 Report
The paper was improved. No other remarks.